# Factors Contributing to the Development of Choroidal Microvasculature Dropout in Glaucoma Suspects and Patients with Glaucoma

**DOI:** 10.3390/jcm13010204

**Published:** 2023-12-29

**Authors:** Hee Jong Shin, Si Eun Oh, Seong Ah Kim, Chan Kee Park, Hae-Young Lopilly Park

**Affiliations:** Department of Ophthalmology, Seoul St. Mary’s Hospital, College of Medicine, The Catholic University of Korea, Seoul 06591, Republic of Korea; shinhj01@naver.com (H.J.S.);

**Keywords:** glaucoma suspect, microvasculature dropout, choroidal microvasculature

## Abstract

We aimed to characterize and compare the occurrence of peripapillary microvasculature dropout (MvD) between glaucoma suspects and patients with glaucoma. In addition, the factors related to the development of parapapillary MvD in glaucoma suspects and patients with glaucoma were investigated. Of a total 150 eyes, 68 eyes of glaucoma suspects and 82 eyes of glaucoma patients were analyzed in this study. Univariate and multivariate logistic regression analyses were used to identify factors associated with MvD development. The classification of glaucoma patients or glaucoma suspects was not significantly associated with MvD development (beta 1.368, 95% CI, 0.718–2.608, *p* = 0.341). In the regression analysis of the glaucoma suspect group, greater axial length (beta 1.520, 95% CI, 1.008–2.291, *p* = 0.046) and baseline cup volume (beta 3.993, 95% CI, 1.292–12.345, *p* = 0.035) among the baseline factors and the slope of ganglion cell–inner plexiform layer (GCIPL) thickness (beta 0.027, 95% CI, 0.072–0.851, *p* = 0.027) and central visual field (VF) progression (beta 7.040, 95% CI, 1.781–16.306, *p* = 0.014) among follow-up factors were significantly associated with MvD development. In the glaucoma group, central VF progression (beta 5.985, 95% CI, 1.474–24.083, *p* = 0.012) and ONH depression (beta 3.765, 95% CI, 1.301–10.895, *p* = 0.014) among follow-up elements were observed as significant factors and the baseline factor had little relationship. MvD appears not only as a result of the progression of axonal loss of RGC in glaucoma but may also be developed due to structural changes and mechanical susceptibility of the ONH associated with baseline characteristics. Analyzing the structural susceptibility of the ONH can predict the occurrence of MvD, which can be helpful in predicting the progression of glaucoma.

## 1. Introduction

Glaucoma is a chronic and progressive eye disease with multifactorial causation, characterized by structural damage to the cellular components of the retina and axonal elements in the optic nerve. The precise pathophysiological mechanisms responsible for glaucoma remain uncertain [1]. Theories include mechanical damage, vascular contribution, biochemical aspects, genetic predisposition, and translaminar pressure gradient [2], but the most widely accepted theories point to mechanical and vascular origins. The mechanical theory hypothesizes that glaucoma occurs because of increased intraocular pressure (IOP), which forces the lamina cribrosa backward and compresses the nerve fibers to disturb axoplasmic flow. The vascular theory attempts to explain glaucoma based on reduced perfusion pressure, vascular dysregulation, or loss of neurovascular coupling [3].

Microvasculature changes in the optic nerve head (ONH) have long been suggested to contribute to glaucoma [4,5]. High-resolution optical coherence tomography (OCT) angiography (OCTA) enables the visualization of the choroidal microvasculature; as OCTA has become common in glaucoma clinics, interest in the finding of peripapillary microvasculature dropout (MvD) has grown. Recent studies have found a correlation of MvD with topographic defects (e.g., retinal nerve fiber layer (RNFL) defects, visual field (VF) defects, and changes in the lamina cribrosa (LC)) [6,7]. As LC changes triggered by elevated intraocular pressure (IOP) are key indicators of glaucomatous ONH, the development of MvD may be related to mechanical changes within and around the ONH. However, MvD has also been detected in healthy (nonglaucomatous) eyes with systemic vascular dysregulation [8]. In glaucoma patients with MvD, glaucoma progression is faster in the presence of abnormal vascular autoregulation, such as disc hemorrhage, cold extremities, migraines, low ocular perfusion pressure, and nocturnal blood pressure dips [9,10,11]. Therefore, vascular instability and insufficiency are also hypothesized to be associated with MvD. In addition, MvD has been observed in patients with compressive optic neuropathy (not directly related to changes in ONH) [12]. This indicates that MvD may also develop because of the reduced metabolic need of damaged axons, which reduces perfusion from the juxtapapillary choroid.

Similar to the pathophysiological mechanism of glaucoma, there is no consensus on the cause of MvD—it is unclear whether it is a mechanical process within the ONH that precedes glaucoma or a secondary change in glaucoma progression. However, we need to elucidate the underlying process of MvD development to understand the clinical significance of this finding on OCTA. We hypothesized that if choroidal MvD results from the mechanical characteristics of the ONH, not from the reduced metabolic demand of retinal ganglion cells, it would precede glaucomatous damage and may be observed in glaucoma suspects. We previously reported that glaucoma suspects with detected MvD showed a greater rate of glaucoma development, which may indicate that it is an ongoing process during glaucoma development and progression [13]. To test our hypothesis, we aimed to characterize and compare the occurrence of MvD in glaucoma suspects and patients with glaucoma. In addition, the factors related to the development of parapapillary MvD in glaucoma suspects and patients with glaucoma were investigated.

## 2. Materials and Methods

### 2.1. Participants

This study was a component of the Catholic Medical Center Glaucoma Progression Study (CMC-GPS) and the Catholic Medical Center Glaucoma Suspect Cohort Study (CMC-GSCS), which began in 2009 at Seoul St. Mary’s Hospital in Seoul, Republic of Korea. This study was approved by the Institutional Review Board of Seoul St. Mary’s Hospital and followed all relevant tenets of the Declaration of Helsinki. We included all consecutive eligible patients who expressed willingness to participate, and each patient provided written informed consent.

All participants underwent complete ophthalmic examinations including a detailed review of medical/ocular records and prescriptions, slit lamp examination, Goldmann applanation tonometry, gonioscopy, measurement of central corneal thickness using ultrasound pachymetry (Tomey Corp., Nagoya, Japan), measurement of axial length using ocular biometry (IOL Master; Carl Zeiss Meditec, Jena, Germany), dilated stereoscopic optic disc examination, red-free fundus photography (Canon, Tokyo, Japan), OCT (Cirrus OCT; Carl Zeiss Meditec and Spectralis OCT; Heidelberg Engineering, Heidelberg, Germany), Humphrey VF examination using the Swedish interactive threshold Standard 24-2 algorithm (Carl Zeiss Meditec), and Heidelberg Retina Tomograph III (HRT III; Heidelberg Engineering, Heidelberg, Germany). Starting in 2017, all patients underwent additional serial OCTA (DRI OCT Triton; Topcon, Tokyo, Japan) examinations and were followed-up every 1–3 months with IOP measurements and optic disc evaluation. Disc photography, VF, OCT, HRT, and OCTA were performed annually. All disc hemorrhages (DHs) occurred during follow-up were recorded. The IOP was recorded at each visit. The mean IOP during the entire follow-up period was calculated by averaging all measurements. IOP fluctuation was calculated as the standard deviation of the mean IOP.

Glaucoma suspects were defined as individuals with clinical findings or risk factors linked to the likelihood of developing glaucoma, including a high intraocular pressure (IOP) repeatedly above 21 mmHg and suspected findings of the optic disc (concentric enlargement of the cup-to-disc ratio or vertical cup-to-disc ratio ≥ 0.6) or thinning of the retinal nerve fiber layer (RNFL), and an open-angle evident on gonioscopy, but with no evidence of optic nerve damage or VF loss. Glaucoma was defined by the presence of a glaucomatous optic disc (with diffuse or localized rim thinning, a notch in the rim or a vertical cup-to-disc ratio ≥ 0.2 than that of the other eye), corresponding reproducible VF loss (a cluster of ≥3 non-edge points on the pattern deviation plot with a probability < 5% of that of the normal population, and with one of these points having a probability of <1%; a pattern standard deviation with a *p* value < 5%, or a Glaucoma Hemifield Test result consistently outside the normal range on two VF examinations) as confirmed by two glaucoma specialists (H.Y.P. and C.K.P.) and an open angle evident on gonioscopy.

The criteria for inclusion in the additional patient population were as follows: a best-corrected visual acuity ≥ 20/40, a spherical refraction within ± 6.0 diopters (D), cylinder correction within ± 3.0 D, at least three dependable VF results (false negatives < 15%, false positives < 15%, and fixation losses < 20%), and a mean deviation (MD) better than −20.00 decibels (dB). The criteria for exclusion were as follows: any retinal disease history, including hypertensive or diabetic retinopathy; eye trauma or surgery history, with the exception of uncomplicated cataract surgery; any optic nerve disease except glaucoma; and history of systemic or neurological conditions that could affect VF test. If laser glaucoma treatment or incisional procedure was performed during follow-up period, only the data obtained before treatment were analyzed. One eye was arbitrarily selected for the study if both eyes met the inclusion criteria.

### 2.2. Definition of Central VF Progression

The patients received two baseline Humphrey 24-2 VF tests every six months for at least 24 months. In the 24-2 VF tests, three or more neighboring points (5% depressed from the normative database) within the 10° region or one or more points (1% depressed from the normative database) with no abnormality outside the central 10° region were defined as the initial paracentral scotoma. To analyze the VF progression, newly appearing VF points within the 10° region that decreased to <5%, <2%, <1%, or <0.5% on the 24-2 VF test from the total deviation plot were evaluated.

### 2.3. HRV (Heart Rate Variability) Assessment

A detailed history of vascular symptoms such as Raynaud’s phenomenon, cold extremities, migraines, and symptoms relevant to orthostatic hypotension was obtained. Patients presenting with any of these symptoms were considered to have vascular symptoms. The participants were asked to avoid fast moving activities such as running for at least 2 h before the HRV test. Testing was performed for 5 min under controlled conditions. Echocardiography (Cardiotens-01; Meditech Ltd., Budapest, Hungary) and a Medicore Heart Rate Analyzer (Model SA-3000P; Medicore, Seoul, Korea) were used to assess the status of the autonomic nervous system and the balance between the sympathetic and parasympathetic systems in terms of heart rate. The standard deviation of the NN interval (SDNN) index was the mean of all normal RR intervals of the standard. This parameter indicates the total regulatory effect on the autonomic circulation. A decrease in SDNN suggests a high tone of heart sympathetic activity and has been found to predict an increased risk of sudden cardiac arrest and vasospastic trait.

### 2.4. OCT Examination

Using Cirrus SD-OCT version 6.0, the peripapillary RNFL thickness was measured using the optic Disc Cube 200 × 200 scan mode, and the ganglion cell–inner plexiform layer (GCIPL) thickness was determined using GCA software (Carl Zeiss Meditec, Dublin, CA, USA) with a macular cube scan. In addition, measurements of ONH parameters, such as rim area and cup volume, were automatically created using a Carl Zeiss Meditec ONH analysis algorithm. The algorithm recognizes the end of the Bruch’s membrane (BM) as the disc edge, and the rim width around the entire circumference of the optic disc is defined by measuring the thickness of the neuroretinal tissue in the optic nerve as it exits through the opening in the BM. Measured within three-dimensional volume, this composes a single-area measure. Detailed protocols for ONH parameters and RNFL/GCIPL thickness have been described previously [14,15].

### 2.5. Analysis of OCT Images for Determining LC Parameters

The Heidelberg Spectralis OCT system furnishes up to 40,000 A-scans/s with a depth resolution of 7 μm in tissues and a transverse resolution of 14 μm in ocular microstructure images. EDI-OCT B-scans around the ONH (6-mm optic cube scans) were acquired using the Spectralis OCT system. Each section was obtained using eye tracking, and an average of at least 35 OCT frames was merged. Images with a quality score > 15 were selected (~65–70 sections per eye). We measured the thickness and depth of the lamina cribrosa (LC) based on the averaged images. Measurements were carried out by two observers (H.Y.P. and S.A.K.) in a blinded manner using the caliper function of the OCT software (version-1.6.1.0). Detailed ONH parameter measurement methods have been announced elsewhere [16]. Eyes with unclear images at the bottom of the LC region were excluded.

LC thickness was determined as the length between the anterior and posterior borders of the hyperreflective region at the bottom of the ONH. Measurements were conducted along a line perpendicular to the reference line, bridging the end of the Bruch’s membrane to the center of the reference line. The LC was decided by measuring the distance from the opening plane of Bruch’s membrane to the level of the anterior LC surface. The average of three values from three separate images throughout the ONH scan was used for each measurement.

### 2.6. OCT-A Examination

The macular and parapapillary regions were imaged by using a commercial swept-source OCT-A device (DRI OCT Triton; Topcon). The central wavelength was 1050 nm, acquisition speed was 100,000 A-scans/s, and axial and transverse resolutions were 7 and 20 µm. Cubes of dimensions 4.5 × 4.5 mm underwent scanning using 320 clusters, each consisting of four repeated B-scans. These scans were centered on both the optic disc and macula within each cube.

En face images, produced through automated layer segmentation of signals from the retinal pigment epithelium to the outer scleral border, were employed to assess the deep-layer parapapillary microvasculature in the specified area. MvD, characterized by focal sectoral capillary dropout within a discernible microvascular network, was identified. MvD was identified based on a dropout width > twofold that of visible juxtapapillary microvessels. Patients without MvD on baseline OCTA but with MvD detected on follow-up OCTAs were defined as eyes with MvD development. Two impartial observers (H.Y.P. and S.A.K.), unaware of the clinical data, detected the MvD. Any discrepancies were resolved by a third observer (C.K.P.). Only images that were sharp (with quality scores exceeding 30 and devoid of motion blurring) were subject to analysis.

### 2.7. Confocal Scanning Laser Ophthalmoscopy to Measure ONH Surface Depression

Optic disc imaging was performed with an HRT of 3 (Heidelberg Engineering). A 3-dimensional topographic image consisting of 384 × 384 × 64 pixels was constructed axially from multiple focal planes along the ONH. An average of three consecutive scans was obtained and aligned to compose a single mean topography for the analysis. An experienced examiner outlined the optic disc margin on the mean topographic images. After delineating the contour line, the software automatically computed various optic disc measurements. The region above the reference plane was designated as the rim, while the region below it was identified as the cup. The reference plane was positioned at 50 μm posterior to the mean retinal height within the 350° to 356° range along the contour line. The cup shape was calculated as a measure of the overall three-dimensional shape of the optic disc. Images of high quality were defined as those exhibiting an image standard deviation below 50 μm, uniform image exposure, and precise centering.

HRT Topographic Change Analysis (TCA, Heidelberg Engineering) was used to analyze serial ONH topography images to detect ONH surface depression. Individual superpixel ONH surface height measurements were compared between the baseline and each follow-up examination using the F test. The combined variability of baseline and follow-up examinations for a specific pixel was contrasted with the within-variability of baseline and follow-up examinations (with an F-test error probability of <5%). If a notable surface depression in the ONH was identified in a superpixel and validated through at least two successive follow-up visits, the superpixel was marked in red on the significance map. The color saturation increased with the magnitude of the surface height change. The characterization of progressive surface depression in the ONH involved three criteria (liberal, moderate, and conservative), based on the extent and depth of ONH surface depression as outlined in the studies conducted by Chauhan and colleagues [17,18]. The liberal criterion required a cluster of ≥0.5% of the disc area and a depth change of ≥20 mm; the moderate criterion required a cluster of ≥1% of the disc area and a depth change of ≥50 mm; and the conservative criterion required a cluster of ≥2% of the disc area and a depth change of ≥100 mm. In the present study, moderate standard was implemented to guarantee reasonable specificity in identifying ONH surface depression. ONH surface depression was defined as the presence of at least three of the four consecutive follow-up examinations.

### 2.8. Statistical Analysis

Two observers (H.Y.P. and S.A.K.) assessed the interobserver reproducibility of ONH parameter measurements in 30 randomly chosen eyes. Intraclass correlation coefficients (ICCs) and 95% confidence intervals (CIs) were then computed. According to Fleiss [19], ICCs ≥ 0.75, 0.40–0.75, and ≤0.4 are excellent, moderate, and poor, respectively. Interobserver differences in MvD identifications were evaluated using к coefficients. Student’s *t*-test and the χ^2^ test were used to compare continuous and categorical variables, respectively. The slopes of the OCT, HRT, and VF parameters were calculated as changes per year using linear regression. Univariate and multivariate logistic regression analyses were used to identify factors associated with MvD development. Independent variables (*p* values < 0.10 in the univariate model were included in the multivariate model. Statistical significance was set at *p* < 0.05. significance. All statistical analyses were performed using the SPSS Statistics software (ver. 16.0; IBM Corp., Armonk, NY, USA).

## 3. Results

A total of 172 eyes from 172 glaucoma suspects or patients were included in this study. Of the 172 eyes, 10 (5.8%) were excluded from the analysis because the disc/RNFL photographs or OCT images were of poor quality or the VF reliability indices were unreliable. An additional 12 eyes (7.0%) were excluded due to poor-quality OCTA images or motion artifacts. Therefore, a total of 150 eyes of 150 glaucoma suspects or glaucoma patients were analyzed in this study. Among the 150 eyes, 68 eyes were from glaucoma suspects and 82 were from patients with glaucoma. ONH measurements showed excellent reproducibility, with ICCs of 0.972–0.987 (95% CI = 0.953–0.998) for the LC thickness and depth. Interobserver agreement in terms of MvD detection was excellent (к = 0.956; 95% CI, 0.913–0.989).

Table 1 presents the baseline characteristics of the participants. Mean age was 56.32 ± 13.91 years old and showed MD of the 24-2 VF test as −3.36 ± 4.43 dB. The average follow-up duration was 8.62 ± 2.20 years. During the follow-up period, 77 (51.3%) eyes developed MvD. Comparison between group who developed MvD and group without MvD, there was a significant difference in the slope of cup volume among OCT parameters, central VF progression, and ONH depression measured using HRT (all *p* < 0.05, Table 2). ONH depression indicates changes in the connective tissues of the ONH such as the lamina cribrosa and non-axonal components of the neuroretinal rim and prelaminar tissue. ONH depression during the HRT was found in 35 (77.8%) patients who developed MvD compared to 13 (35.1%) patients who did not develop MvD (*p* < 0.001). There was no significant difference in the frequency of MvD occurrence between patients with glaucoma (*n* = 45, 54.9%) and glaucoma suspects (*n* = 32, 47.1%) (*p* = 0.215). In the total regression analysis of factors associated with the development of MvD, the classification of glaucoma patients or glaucoma suspects was not significantly associated with MvD development (beta 1.368, 95% CI, 0.718–2.608, *p* = 0.341; Table 3).

Thinner baseline GCIPL thickness, presence of central VF progression, and slope of cup shape measured using HRT were significantly associated with MvD development in multivariate analysis in all participants (all *p* < 0.05, Table 3). Vascular symptoms or SDNN in the HRV test were not significantly related to MvD occurrence, and it was assumed that systematic blood flow dysregulation had no effect on the occurrence of MvD in the present cohort.

In the glaucoma suspect group, MvD developed in 32 (47.1%) of 68 eyes. A comparison between glaucoma suspects who developed MvD and those who did not is shown in Table 4. Glaucoma suspects who developed MvD had a significantly longer axial length, thinner central corneal thickness, higher baseline IOP, thinner baseline rim area, greater baseline cup volume, thinner baseline average RNFL thickness, thinner baseline macular GCIPL thickness, and a greater frequency of developing central scotoma on VF (all *p* < 0.05, Table 4). As a result, there were significantly more cases that converted to glaucoma in the glaucoma suspect group that developed MvD (*p* = 0.021, Table 4). In the regression analysis to analyze factors associated with MvD development in the glaucoma suspect group, greater axial length and baseline cup volume among the baseline factors were identified as significant factors associated with the occurrence of MvD (all *p* < 0.05, Table 5), and the slope of the GCIPL thickness and central progression of VF among the follow-up factors were significantly associated with MvD development (all *p* < 0.05, Table 6).

In contrast, MvD developed in 45 (54.9%) of 82 eyes with glaucoma. When analyzed by dividing them into MvD and non-MvD development groups, there were significant differences in age, cup volume slope, central progression of the VF, and ONH surface depression using HRT (all *p* < 0.05, Table 7). In the regression analysis, the central progression of VF and ONH surface depression using HRT among follow-up elements was observed as a significant factor associated with MvD development (all *p* < 0.05, Table 8). The baseline factor had little relationship with the occurrence of MvD in the glaucoma group, unlike in the glaucoma suspect group (Table 9).

Representative cases are shown in Figure 1 and Figure 2. In one case, a 71-year-old woman with a glaucoma suspect in her left eye. Glaucomatous damage presented as an enlargement of vertical cup-to-disc ratio and inferotemporal RNFL thinning without noticeable VF defect. A choroidal map of OCT-A imaging revealed no parapapillary MvD. In the follow-up examination conducted two years later, MvD was newly observed, and a localized inferotemporal RNFL defect was detected. A new paracentral scotoma was found in the VF examination. In the other case, a 47-year-old woman with NTG had a localized inferotemporal RNFL defect and corresponding isolated superior arcuate scotoma in her left eye. No MvD was observed at a choroidal map of OCT-A image. In the follow-up examination conducted three years later, new MvD and a widening in RNFL defect were observed. Central VF progression was also detected.

## 4. Discussion

In this study, we aimed to determine whether the development of parapapillary choroidal MvD precedes the loss of axons and accompanying tissues or follows diminished vascular need due to axon loss. In the glaucoma suspect group, which did not have clinically apparent axonal loss due to glaucoma, baseline factors related to the mechanical vulnerability of the ONH and myopia were associated with MvD development. Baseline factors, such as longer axial length, thinner central corneal thickness, higher baseline IOP, and larger cup volume at baseline, which may indicate greater structural vulnerability, were associated with MvD development in the glaucoma suspect group. This finding suggests that the presence of MvD may be an indicator of early ongoing processes occurring within and around the ONH preceding apparent axonal loss. In contrast, baseline factors were not associated with MvD development in glaucomatous eyes, and only follow-up factors, such as central VF progression and ONH surface depression using HRT, were significantly associated with MvD development. Both central VF progression and macular GCIPL thinning were also significantly associated with MvD development in glaucoma suspects. Therefore, MvD is associated with glaucomatous damage, particularly in the central macular region. This is consistent with previous studies showing that MvD develops preferentially at the inferotemporal sector, which is close to the neuroretinal rim that receives the papillomacular bundle, and consequently contributes to development of parafoveal scotoma [9,20]. We suggest that MvD detection in glaucoma suspects may indicate susceptibility to glaucoma and the occurrence of mechanical changes in the ONH preceding axonal loss, and that these eyes are at risk of developing glaucoma. When MvD is detected in glaucomatous eyes, progression and ongoing axonal loss in the ONH or central macular region should be evaluated, and MvD occurrence may be a secondary finding accompanying glaucoma progression.

A peripapillary choroidal microvasculature defect (MvD) is a new ocular finding recently revealed using OCT-A that describes the reduction or loss of small blood vessels in the choroid, a layer of tissue located beneath the retina that supplies oxygen and nutrients to the retina and optic nerve head [7,21,22]. MvD can be seen within the beta (β)-zone of parapapillary atrophy (PPA) on choroidal vessel density maps of the ONH generated using OCT-A. This defect is observed in patients with glaucoma and is believed to play a role in disease development and progression. The exact mechanism by which peripapillary choroidal MVD develops in patients with glaucoma is not fully understood; however, MvD may form during the glaucomatous process as it is found more frequently in glaucomatous eyes. Open-angle glaucoma (OAG) eyes with LC defect have lower parapapillary vessel densities [23]. A larger β-PPA and the presence of LC defect were associated with choroidal MvD in OAG eyes [22]. The current study also showed results similar to those of previous studies as rim area and cup volume related to LC were found to be related to the occurrence of MvD.

In our previous study, a higher baseline IOP was significantly associated with the extent of MvD in glaucoma patients [24]. Higher baseline IOP was also shown to be a significant factor of MvD development in glaucoma suspects in this study. This finding is consistent with the mechanical theory that glaucoma occurs as LC deformation due to a relatively high intraocular pressure. Myopic eyes or eyes with thinner central cornea are prone to LC changes, even under normal IOP, and both are well-known factors that contribute to glaucoma development. Taken together, the baseline factors associated with the development of MvD in glaucoma suspects indicate that greater structural vulnerability is associated with MvD development in the glaucoma suspect group. This suggests that the presence of MvD could be an indicator of early ongoing processes occurring within and around the ONH preceding apparent axonal loss.

TCA parameters, including ONH surface depression measured using HRT, are known to show high sensitivity and specificity in viewing optic disc changes and are reported to sensitively detect early disc progression [17,25,26]. In this study, there was a significant difference in ONH depression in glaucoma patients due to the presence of MvD, and it was found to be a significantly associated factor in the multivariate regression of MvD occurrence in the glaucoma group. However, RNFL thickness was not significantly associated with the MvD development. Similarly, a previous study has shown that corneal hysteresis has a significant correlation with ONH surface depression measured using HRT, but not with RNFL thinning [27]. This study and our study are consistent with the results of a previous study showing that ONH surface depression occurs before RNFL thinning in a significant proportion of patients with glaucoma [28]. ONH depression may reflect changes in both LC and prelaminar tissue and rim loss owing to RNFL thinning. However, ONH surface depression was significantly associated with MvD development in glaucomatous eyes but not in glaucoma suspects. Although ONH depression caused by HRT could result from both ONH changes and rim loss, our findings indicate that ongoing axonal loss at the ONH was likely to be primarily associated with the development of MvD, which may be a secondary finding accompanying glaucoma progression.

Interestingly, central VF progression and its associated GCIPL thickness slope were identified as significant elements among the follow-up factors in patients who developed MvD. Systemic factors such as migraine, Raynaud’s phenomenon, and hypotension are considered as important risk factors for central scotoma [29,30]. Moreover, in our previous study, decreased deep macular vessel density was an independent risk factor for central scotoma [31]. It is not possible to completely exclude vascular factors in the occurrence of MvD. According to a recent study, vascular factors are intimately associated with MvD [9,10,11]. Nevertheless, the occurrence of MvD is explained by mechanical reasons, and in patients with MvD, vascular dysregulation additionally worsens blood flow insufficiency in certain areas and causes central VF progression, leading to the conclusion that MvD is associated with vascular factors in regions with mechanical changes at the level of the LC or peripapillary sclera. In this study, underlying DM, HTN, SDNN of the HRV test, and vascular symptoms related to the systemic vascular condition of patients in the baseline state did not affect the occurrence of MvD; therefore, mechanical and structural changes are mainly associated with the development of MvD, and vascular insufficiency may affect blood flow in this region. This additive local reduction in blood flow at the structures related to MvD with mechanical changes may contribute to the function of the macular region because MvD is thought to be closely related to the progression of central VF.

Our study has several limitations. First, OCT-A imaging is an emerging technique, and the retinal vessel signals evident on en face deep-layer OCTA images make it difficult to precisely define MvD boundaries. Therefore, only eyes with clear dropouts were considered as having MvD. However, the fact that MvD is located within the zone of the PPA, which has few superficial retinal vessels and where the signals may not be blocked by the disc rim, reduces the risk of artifacts. Second, the majority of patients with glaucoma had normal-tension glaucoma; therefore, there is a possibility of a selection bias that included only patients who were not weak from the beginning when selecting glaucomatous eyes without MvD. In this case, the baseline factors may not have been identified in the association analysis. Glaucoma suspects were also mostly normal-tension glaucoma suspects with normal range of IOP; therefore, differences in factors associated with the development of MvD may be meaningful in managing these patients. Third, the vascular symptoms and HRV test were used in this study to evaluate the autonomic dysfunction, but these results alone may be insufficient to figure out systemic blood flow dysregulation. Kurysheva et al. reported that the cold provocation test should be performed to more clearly observe changes in HRV parameters, including SDNN, in glaucoma patients [32]. In addition, our previous study also reported that central visual field progression in NTG is associated with an autonomic dysfunction [33]. Therefore, it may be difficult to say that the occurrence of MvD associated with central visual field progression is entirely unrelated to blood flow instability. Finally, the functional damage of RGC was likely in progress even in patients selected as glaucoma suspects, even if the RNFL defect was undetectable. Therefore, our results may be a combined effect of both ONH changes and the damage process and related reduction in vascular need. However, these findings require further investigation. Nevertheless, it was confirmed that various factors influenced the occurrence of MvD in patients. This is the first study to provide a solid explanation for the development of MvD, showing that it can occur primarily because of changes within the ONH.

## 5. Conclusions

In summary, MvD appears not only as a result of the progression of axonal loss of RGC in glaucoma but may also be developed by structural changes and mechanical susceptibility of the ONH associated with baseline characteristics, including myopia, thin central corneal thickness, and higher baseline IOP. The progression of ONH changes and damage in the central macular region also resulted in the development of MvD in both glaucoma suspects and patients with glaucoma, which could be identified by GCIPL thinning and central VF progression. Therefore, analyzing the structural susceptibility of the ONH can predict the occurrence of MvD, which can be helpful in predicting the progression of glaucoma. However, longitudinal studies are required to confirm these findings.

## Figures and Tables

**Figure 1 jcm-13-00204-f001:**
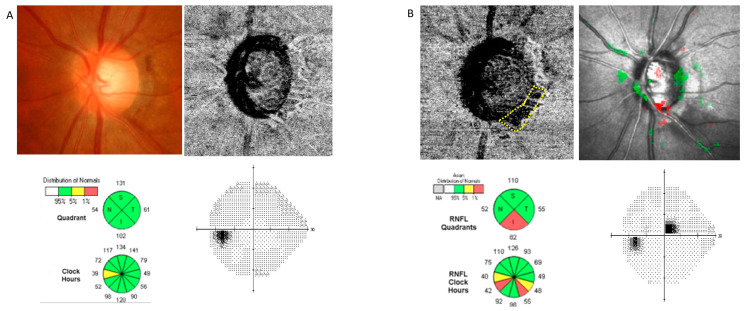
A representative case of a patient with glaucoma suspect who developed microvasculature dropout (MvD). The patient’s axial length was 24.36 mm, initial IOP was 11 mmHg, and IOP fluctuation was measured between 11 and 14 mmHg during the follow-up. (**A**) Glaucomatous damage presented as an enlargement in vertical cup-to-disc ratio and inferotemporal retinal nerve fiber layer (RNFL) thinning without noticeable visual field (VF) defect. A choroidal map of optical coherence tomography angiography (OCT-A) imaging revealed no parapapillary MvD. (**B**) In the follow-up examination conducted two years later, MvD was newly observed (displayed by a yellow dotted line), and localized inferotemporal RNFL defect was detected. A new paracentral scotoma was found in the VF examination.

**Figure 2 jcm-13-00204-f002:**
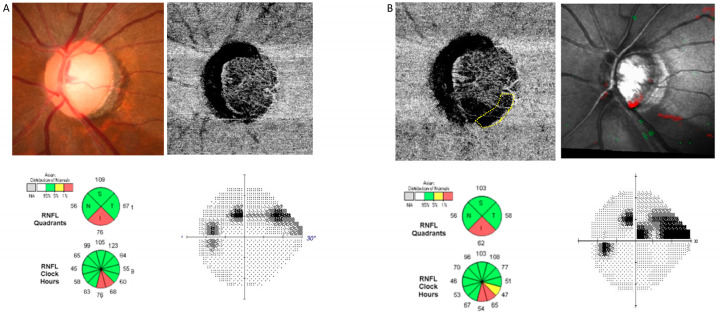
A representative case of a patient with normal tension glaucoma (NTG) who developed microvasculature dropout (MvD). The patient’s axial length was 24.80 mm, initial IOP was 15 mmHg, and IOP fluctuation was measured between 11 and 15 mmHg during the follow-up. (**A**) Glaucomatous damage presented as a localized inferotemporal retinal nerve fiber layer (RNFL) defect with corresponding isolated superior arcuate scotoma. A choroidal map of optical coherence tomography angiography (OCT-A) imaging revealed no parapapillary MvD. (**B**) In the follow-up examination conducted three years later, new MvD (displayed by a yellow dotted line) and a widening in RNFL defect were observed. Central visual field (VF) progression was also detected.

**Table 1 jcm-13-00204-t001:** Baseline demographics and ocular characteristics of 82 eyes of 82 glaucoma patients and 68 eyes of 68 glaucoma suspects.

Demographics	Total (*n* = 150)	Glaucoma Suspect (*n* = 68)	Glaucoma (*n* = 82)
Age at diagnosis, y	56.32 ± 13.91	54.48 ± 12.77	57.83 ± 12.58
Female, no. (%)	62 (41.3%)	35 (73.5%)	27 (32.9%)
Systemic demographics	7 (4.7%)	4 (5.9%)	3 (3.7%)
Medication of DM, no. (%)	39 (26.0%)	17 (25.0%)	22 (26.8%)
Medication of HTN, no. (%)	21 (14.0%)	2 (2.9%)	19 (23.2%)
Vascular symptoms, no. (%)	37.36 ± 25.20	36.96 ± 25.11	37.96 ± 23.17
SDNN of heart rate variability test			
Ocular demographics	24.26 ± 4.34	24.96 ± 4.56	24.93 ± 5.11
Axial length, mm	540.13 ± 39.45	552.3 ± 39.52	531.26 ± 38.60
Central corneal thickness, μm	35 (23.3%)	0	35 (42.7%)
Presence of DH, no. (%)			
IOP parameters	14.89 ± 2.87	15.08 ± 2.77	14.73 ± 2.95
Baseline IOP, mmHg	14.45 ± 2.79	14.49 ± 2.84	14.40 ± 3.01
IOP during follow-up, mmHg	1.91 ± 2.38	1.59 ± 2.32	2.00 ± 2.37
IOP fluctuation during follow-up, mmHg			
OCT parameters			
Rim area, mm^2^	0.89 ± 0.25	0.95 ± 0.25	0.88 ± 0.24
Cup volume, mm^3^	0.49 ± 0.26	0.49 ± 0.23	0.54 ± 0.27
Average pRNFL thickness, μm	80.37 ± 12.99	87.12 ± 12.84	74.94 ± 13.01
Average mGC/IPL thickness, μm	73.07 ± 9.20	75.65 ± 9.18	70.98 ± 9.62
VF parameters			
MD, dB	−3.36 ± 4.43	−1.60 ± 2.78	−4.90 ± 5.01
PSD, dB	3.82 ± 3.76	2.14 ± 3.03	5.19 ± 4.40
Disc parameters by HRT			
Disc area, mm^2^	2.23 ± 0.52	2.36 ± 0.48	2.13 ± 0.47
Cup shape measures	−0.09 ± 0.07	−0.09 ± 0.07	−0.10 ± 0.06
Cup depth	0.64 ± 0.17	0.63 ±0.15	0.64 ± 0.16
Measured ONH parameters			
LCD, μm	436.32 ± 118.45	427.54 ± 119.32	451.27 ± 120.44
LCT, μm	208.17 ± 47.60	208.58 ± 48.16	207.47 ± 46.23
MvD on OCT-A, no. (%)	77 (51.3%)	32 (47.1%)	45 (54.9%)
Follow-up duration, y	8.62 ± 2.20	8.60 ± 2.19	9.09 ± 2.18

DM: diabetes mellitus; HTN: systemic hypertension; OCT: optical coherence tomography; pRNFL: peripapillary retinal nerve fiber layer; mGC/IPL: macular ganglion cell–inner plexiform layer; IOP: intraocular pressure; VF: visual field; MD: mean deviation; PSD: pattern standard deviation; dB: decibel; DH: disc hemorrhage; HRT: Heidelberg retinal tomograph; ONH: optic nerve head; LCD: lamina cribrosa depth; LCT: lamina cribrosa thickness; MvD: microvasculature dropout; OCT-A: optical coherence tomography angiography. Data are mean ± standard deviation unless otherwise indicated.

**Table 2 jcm-13-00204-t002:** Comparison between eyes that developed or did not develop microvasculature dropout on optical coherence tomography angiography in a total of 150 eyes.

Variables	MvD Development (*n* = 77)	No MvD Development (*n* = 73)	*p* Value
Demographics			
Age at diagnosis, y	54.33 ± 14.52	58.41 ± 12.99	0.072 *
Female, no. (%)	33 (42.9%)	29 (39.7%)	0.135 ^†^
Systemic demographics			
Medication of DM, no. (%)	2 (2.6%)	5 (3.8%)	0.200 ^†^
Medication of HTN, no. (%)	22 (28.5%)	17 (23.3%)	0.291 ^†^
Vascular symptoms, no. (%)	13 (16.9%)	8 (11.0%)	0.209 ^†^
SDNN of heart rate variability test	38.92 ± 27.20	35.55 ± 23.08	0.516 *
Ocular demographics			
Axial length, mm	24.53 ± 4.29	23.97 ± 4.38	0.706 *
Central corneal thickness, μm	536.56 ± 37.66	543.51 ± 41.02	0.756 *
Presence of DH, no. (%)	22 (48.9%)	13 (35.1%)	0.152 ^†^
IOP parameters			
Baseline IOP, mmHg	15.02 ± 3.04	14.74 ± 2.68	0.292 *
IOP during follow-up, mmHg	14.51 ± 2.64	14.39 ± 2.96	0.292 *
IOP fluctuation during follow-up, mmHg	1.73 ± 2.55	2.10 ± 2.18	0.160 *
OCT parameters			
Baseline rim area, mm^2^	0.85 ± 0.21	0.95 ± 0.28	0.212 *
Slope of rim area, mm^2^/y	−0.06 ± 0.12	−0.04 ± 0.13	0.768 *
Baseline cup volume, mm^3^	0.50 ± 0.27	0.48 ± 0.25	0.350 *
Slope of cup volume, mm^3^/y	0.04 ± 0.08	0.03 ± 0.07	**0.010 ***
Baseline average pRNFL thickness, μm	78.51 ± 11.95	82.33 ± 13.82	0.073 *
Slope of RNFL thickness, μm/y	−0.54 ± 0.59	−0.46 ± 0.70	0.438 *
Baseline average mGC/IPL thickness, μm	70.96 ± 8.55	75.35 ± 9.39	**0.003** *
Slope of mGC/IPL thickness	−0.12 ± 1.36	−0.25 ± 0.71	0.496 *
VF parameters			
Baseline MD, dB	−3.63 ± 4.61	−3.08 ± 4.26	0.447 *
Slope of MD, dB/y	−0.25 ± 1.10	−0.04 ± 0.54	0.134 *
Baseline PSD, dB	4.36 ± 4.10	3.25 ± 3.30	0.071 *
Slope of PSD, dB/y	0.61 ± 2.57	0.13 ± 0.39	0.116 *
Central progression, *n* (%)	31 (40.3%)	4 (5.6%)	**<0.001** ^†^
Disc parameters using HRT			
HRT ONH depression, *n* (%)	51 (66.2%)	30 (41.1%)	**0.002** ^†^
Disc area, mm^2^	2.13 ± 0.49	2.13 ± 0.45	0.410 *
Cup shape measures	−0.11 ± 0.07	−0.10 ± 0.10	0.868 *
Slope of cup shape measures	0.01 ± 0.03	−0.00 ± 0.04	0.063 *
Cup depth	0.62 ± 0.11	0.64 ± 0.16	0.493 *
Slope of cup depth	0.01 ± 0.03	0.00 ± 0.03	0.880 *
Measured ONH parameters			
LCD, μm	440.80 ± 114.51	430.45 ± 124.89	0.700 *
Slope of LCD, μm/y	0.69 ± 3.40	0.67 ± 3.22	0.973 *
LCT, μm	203.19 ± 37.00	214.70 ± 58.67	0.284 *
Slope of LCT, μm/y	−3.08 ± 6.84	−0.96 ± 8.00	0.194 *
Follow-up duration, y	8.57 ± 2.00	8.67 ± 2.41	0.783 *

DM: diabetes mellitus; HTN: systemic hypertension; OCT: optical coherence tomography; pRNFL: peripapillary retinal nerve fiber layer; mGC/IPL: macular ganglion cell–inner plexiform layer; IOP: intraocular pressure; VF: visual field; MD: mean deviation; PSD: pattern standard deviation; dB: decibel; DH: disc hemorrhage; HRT: Heidelberg retinal tomograph; ONH: optic nerve head; LCD: lamina cribrosa depth; LCT: lamina cribrosa thickness; MvD: microvasculature dropout. Data are mean ± standard deviation unless otherwise indicated. * Student’s *t*-test. ^†^ Chi square test. Data are mean ± standard deviation unless otherwise indicated. Factors with statistical significance are shown in bold.

**Table 3 jcm-13-00204-t003:** Factors associated with MvD development in a total of 150 eyes.

Variables	Univariate		Multivariate	
	Beta (95% CI)	*p* Value	Beta (95% CI)	*p* Value
Age at diagnosis	0.979 (0.956–1.002)	0.075	0.999 (0.958–1.043)	**0.980**
Diagnosis of glaucoma suspect vs. glaucoma	1.368 (0.718–2.608)	0.341		
Female	1.526 (0.794–2.933)	0.205		
Medication of DM	0.363 (0.068–1.931)	0.235		
Medication of HTN	1.318 (0.632–2.746)	0.461		
Vascular symptoms	1.650 (0.641–4.250)	0.299		
SDNN of heart rate variability test	1.005 (0.989–1.022)	0.513		
Axial length	1.032 (0.954–1.115)	0.433		
Central corneal thickness	1.005 (0.996–1.013)	0.281		
Baseline IOP	1.036 (0.926–1.159)	0.540		
IOP during follow-up	1.014 (0.904–1.138)	0.810		
IOP fluctuation during follow-up	0.935 (0.812–1.077)	0.351		
Baseline rim area	0.181 (0.045–0.733)	0.017	1.242 (0.255–2.233)	0.242
Slope of rim area	0.266 (0.019–3.737)	0.326		
Baseline cup volume	1.453 (0.422–5.005)	0.554		
Slope of cup volume	2.340 (0.230–21.729)	0.183		
Presence of DH	1.766 (0.723–4.312)	0.212		
Baseline average pRNFL thickness	0.977 (0.952–1.002)	0.075	0.996 (0.921–1.077)	0.914
Slope of RNFL thickness	0.819 (0.496–1.353)	0.436		
Baseline average mGC/IPL thickness	0.946 (0.911–0.983)	0.005	0.895 (0.813–0.987)	0.026
Slope of mGC/IPL thickness	0.899 (0.661–1.224)	0.500		
Baseline MD of VF	0.972 (0.903–1.046)	0.446		
Slope of MD	0.733 (0.482–1.116)	0.148		
Baseline PSD of VF	1.086 (0.991–1.190)	0.076	1.222 (0.978–1.528)	0.078
Slope of PSD	1.876 (1.013–3.474)	0.045	2.088 (0.838–5.205)	0.114
Central progression	1.087 (1.029–2.640)	<0.001	1.370 (1.032–1.594)	**0.008**
HRT ONH depression	2.812 (1.448–5.460)	0.002	0.979 (0.956–1.002)	0.182
HRT depression location	2.250 (0.891–5.685)	0.086	0.829 (0.223–3.088)	0.780
Disc area	1.300 (0.699–2.417)	0.408		
Baseline cup shape measures	1.429 (0.022–92.698)	0.867		
Slope of cup shape measures	4.270 (0.564–13.723)	0.066	2.348 (1.868–4.242)	**0.038**
Baseline cup depth	0.191 (0.026–1.384)	0.101		
Slope of cup depth	0.742 (0.016–3.697)	0.879		
Baseline LCD	1.001 (0.997–1.005)	0.696		
Slope of LCD	1.002 (0.877–1.146)	0.973		
Baseline LCT	0.995 (0.985–1.004)	0.284		
Slope of LCT	0.961 (0.904–1.021)	0.197		
Follow-up duration	0.980 (0.847–1.133)	0.781		

Beta: odds ratio; CI: confidence interval; DM: diabetes mellitus; HTN: systemic hypertension; OCT: optical coherence tomography; pRNFL: peripapillary retinal nerve fiber layer; mGC/IPL: macular ganglion cell–inner plexiform layer; IOP: intraocular pressure; VF: visual field; MD: mean deviation; PSD: pattern standard deviation; dB: decibel; DH: disc hemorrhage; HRT: Heidelberg retinal tomograph; ONH: optic nerve head; LCD: lamina cribrosa depth; LCT: lamina cribrosa thickness; MvD: microvasculature dropout. Data are mean ± standard deviation unless otherwise indicated. Factors with *p* < 0.1 in univariate analysis were included in multivariate analysis. Data are mean ± standard deviation unless otherwise indicated. Factors with statistical significance are shown in bold.

**Table 4 jcm-13-00204-t004:** Comparison between 68 glaucoma suspect eyes that developed or did not develop microvasculature dropout on optical coherence tomography angiography.

Variables	MvD Development (*n* = 32)	No MvD Development (*n* = 36)	*p* Value
Demographics			
Age at diagnosis, y	53.59 ± 14.89	55.31 ± 13.83	0.626 *
Female, no. (%)	15 (46.9%)	20 (55.6%)	0.902 ^†^
Systemic demographics			
Medication of DM, no. (%)	2 (6.2%)	2 (5.6%)	0.647 ^†^
Medication of HTN, no. (%)	9 (28.1%)	8 (22.2%)	0.389 ^†^
Vascular symptoms, no. (%)	1 (3.1%)	1 (2.8%)	0.723 ^†^
SDNN of heart rate variability test	34.64 ± 23.56	37.14 ± 30.67	0.811 *
Ocular demographics			
Axial length, mm	25.40 ± 1.77	24.51 ± 1.84	**0.046** *
Central corneal thickness, μm	540.69 ± 43.35	562.21 ± 41.20	**0.040** *
Presence of DH, no. (%)	0	0	
IOP parameters			
Baseline IOP, mmHg	15.87 ± 3.28	14.36 ± 2.11	**0.026** *
IOP during follow-up, mmHg	14.59 ± 2.69	14.44 ± 1.79	0.787 *
IOP fluctuation during follow-up, mmHg	1.59 ± 1.79	1.47 ± 1.55	0.766 *
OCT parameters			
Baseline rim area, mm^2^	0.86 ± 0.15	0.99 ± 0.28	**0.028** *
Slope of rim area, mm^2^/y	−0.03 ± 0.11	−0.00 ± 0.14	0.420 *
Baseline cup volume, mm^3^	0.62 ± 0.27	0.48 ± 0.18	**0.012** *
Slope of cup volume, mm^3^/y	0.01 ± 0.08	0.02 ± 0.08	0.505 *
Baseline average pRNFL thickness, μm	84.31 ± 7.45	89.25 ± 8.95	**0.017** *
Slope of RNFL thickness, μm/y	−0.49 ± 0.62	−0.34 ± 0.73	0.356 *
Baseline average mGC/IPL thickness, μm	73.46 ± 6.82	77.71 ± 8.27	**0.027** *
Slope of mGC/IPL thickness	−0.55 ± 0.82	−0.10 ± 0.50	**0.009** *
VF parameters			
Baseline MD of SAP, dB	−1.27 ± 1.77	−1.71 ± 2.05	0.455 *
Slope of MD, dB/y	−0.14 ± 0.55	−0.04 ± 0.55	0.438 *
Baseline PSD of SAP, dB	2.09 ± 1.22	2.24 ± 1.23	0.630 *
Slope of PSD, dB/y	0.09 ± 0.43	0.01 ± 0.23	0.316 *
Presence of central scotoma, *n* (%)	11 (34.4%)	1 (2.8%)	**0.001** ^†^
Disc parameters using HRT			
HRT ONH depression, *n* (%)	16 (50.0%)	17 (47.2%)	0.506 ^†^
Disc area, mm^2^	2.45 ± 0.52	2.26 ± 0.57	0.172 *
Cup shape measures	−0.08 ± 0.05	−0.09 ± 0.07	0.398 *
Slope of cup shape measures	0.01 ± 0.04	−0.00 ± 0.04	0.345 *
Cup depth	0.61 ± 0.18	0.68 ± 0.21	0.152 *
Slope of cup depth	0.04 ± 0.10	0.05 ± 0.13	0.764 *
Measured ONH parameters			
LCD, μm	461.21 ± 154.24	442.56 ± 138.47	0.730 *
Slope of LCD, μm/y	0.97 ± 3.28	0.81 ± 2.13	0.875 *
LCT, μm	208.00 ± 41.97	207.00 ± 72.72	0.964 *
Slope of LCT, μm/y	−3.10 ± 7.55	−2.07 ± 5.20	0.665 *
Conversion to glaucoma, *n* (%)	11 (34.4%)	4 (11.1%)	**0.021** ^†^
Follow-up duration, y	7.41 ± 1.29	7.50 ± 1.21	0.758 *

DM: diabetes mellitus; HTN: systemic hypertension; OCT: optical coherence tomography; pRNFL: peripapillary retinal nerve fiber layer; mGC/IPL: macular ganglion cell–inner plexiform layer; IOP: intraocular pressure; VF: visual field; MD: mean deviation; PSD: pattern standard deviation; dB: decibel; SAP: standard automated perimetry; DH: disc hemorrhage; HRT: Heidelberg retinal tomograph; ONH: optic nerve head; LCD: lamina cribrosa depth; LCT: lamina cribrosa thickness; MvD: microvasculature dropout. Data are mean ± standard deviation unless otherwise indicated. * Student’s *t*-test. ^†^ Chi square test. Data are mean ± standard deviation unless otherwise indicated. Factors with statistical significance are shown in bold.

**Table 5 jcm-13-00204-t005:** Baseline factors associated with MvD development in 68 glaucoma suspect eyes.

Variables	Univariate		Multivariate	
	Beta (95% CI)	*p* Value	Beta (95% CI)	*p* Value
Age at diagnosis	0.991 (0.959–1.026)	0.619		
Female	0.900 (0.771–1.060)	0.702		
Medication of DM	1.133 (0.150–8.548)	0.903		
Medication of HTN	1.370 (0.456–4.117)	0.575		
Vascular symptoms	1.129 (0.068–1.822)	0.933		
SDNN of heart rate variability test	0.966 (0.969–1.025)	0.802		
Axial length	1.322 (1.003–1.752)	**0.050**	1.520 (1.008–2.291)	**0.046**
Central corneal thickness	0.912 (0.823–0.998)	**0.045**	1.000 (0.980–1.019)	0.963
Baseline IOP	1.237 (1.016–1.505)	**0.034**	1.232 (0.930–1.631)	0.145
Baseline rim area	0.074 (0.006–0.869)	**0.038**	0.559 (0.031–1.044)	0.693
Baseline cup volume	1.669 (1.006–16.674)	**0.017**	3.993 (1.292–12.345)	**0.035**
Baseline average pRNFL thickness	0.928 (0.871–0.990)	**0.023**	0.941 (0.863–1.027)	0.174
Baseline average mGC/IPL thickness,	0.927 (0.864–0.994)	**0.034**	0.966 (0.885–1.054)	0.437
Baseline MD of VF	1.131 (0.873–1.464)	0.351		
Baseline PSD of VF	0.905 (0.606–1.351)	0.625		
Disc area	1.877 (0.758–4.648)	0.173		
Cup shape measures	2.608 (0.013–6.983)	0.383		
Cup depth	0.158 (0.013–1.994)	0.154		
LCD	1.001 (0.996–1.006)	0.719		
LCT	1.000 (0.988–1.013)	0.963		

Beta: odds ratio; CI: confidence interval; DM: diabetes mellitus; HTN: systemic hypertension; OCT: optical coherence tomography; pRNFL: peripapillary retinal nerve fiber layer; mGC/IPL: macular ganglion cell–inner plexiform layer; IOP: intraocular pressure; VF: visual field; MD: mean deviation; PSD: pattern standard deviation; dB: decibel; DH: disc hemorrhage; HRT: Heidelberg retinal tomograph; LCD: lamina cribrosa depth; LCT: lamina cribrosa thickness; MvD: microvasculature dropout. Data are mean ± standard deviation unless otherwise indicated. Factors with *p* < 0.1 in univariate analysis were included in multivariate analysis. Data are mean ± standard deviation unless otherwise indicated. Factors with statistical significance are shown in bold.

**Table 6 jcm-13-00204-t006:** Follow-up factors associated with MvD development in 68 glaucoma suspect eyes.

Variables	Univariate		Multivariate	
	Beta (95% CI)	*p* Value	Beta (95% CI)	*p* Value
IOP during follow-up	1.030 (0.832–1.276)	0.783		
IOP fluctuation during follow-up	1.046 (0.783–1.396)	0.762		
Slope of rim area	0.190 (0.003–1.055)	0.418		
Slope of cup volume	0.122 (0.000–5.657)	0.502		
Slope of RNFL thickness	0.712 (0.349–1.455)	0.352		
Slope of mGC/IPL thickness	0.286 (0.096–0.855)	**0.025**	0.027 (0.072–0.851)	**0.027**
Slope of MD	0.701 (0.289–1.703)	0.433		
Slope of PSD	2.143 (0.473–9.700)	0.322		
Central progression	1.333 (2.206–5.234)	**0.007**	7.040 (1.781–16.306)	**0.014**
HRT ONH depression	1.118 (0.431–2.899)	0.819		
Slope of cup shape measures	2.232 (0.003–5.429)	0.341		
Slope of cup depth	0.520 (0.008–3.445)	0.760		
Slope of LCD	1.023 (0.779–1.343)	0.869		
Slope of LCT	0.974 (0.866–1.094)	0.653		
Glaucoma conversion	4.190 (1.177–14.920)	**0.027**	1.686 (0.364–7.813)	0.504
Follow-up duration	0.940 (0.638–1.394)	0.754		

Beta: odds ratio; CI: confidence interval; DM: diabetes mellitus; HTN: systemic hypertension; OCT: optical coherence tomography; pRNFL: peripapillary retinal nerve fiber layer; mGC/IPL: macular ganglion cell–inner plexiform layer; IOP: intraocular pressure; VF: visual field; MD: mean deviation; PSD: pattern standard deviation; dB: decibel; DH: disc hemorrhage; HRT: Heidelberg retinal tomograph; ONH: optic nerve head; LCD: lamina cribrosa depth; LCT: lamina cribrosa thickness; MvD: microvasculature dropout. Data are mean ± standard deviation unless otherwise indicated. Factors with *p* < 0.1 in univariate analysis were included in multivariate analysis. Data are mean ± standard deviation unless otherwise indicated. Factors with statistical significance are shown in bold.

**Table 7 jcm-13-00204-t007:** Comparison between 82 glaucomatous eyes that developed or did not develop microvasculature dropout on optical coherence tomography angiography.

Variables	MvD Development (*n* = 45)	No MvD Development (*n* = 37)	*p* Value
Demographics			
Age at diagnosis, y	54.86 ± 14.41	61.43 ± 11.51	**0.028** *
Female, no. (%)	18 (40.0%)	9 (24.3%)	0.102 ^†^
Systemic demographics			
Medication of DM, no. (%)	0 (0%)	3 (8.1%)	0.088 ^†^
Medication of HTN, no. (%)	13 (28.9%)	9 (24.3%)	0.417 ^†^
Vascular symptoms, no. (%)	12 (26.7%)	7 (18.9%)	0.288 ^†^
SDNN of heart rate variability test	40.50 ± 28.55	34.83 ± 19.29	0.350 *
Ocular demographics			
Axial length, mm	23.92 ± 5.36	23.45 ± 5.88	0.706 *
Central corneal thickness, μm	551.61 ± 41.41	554.27 ± 38.13	0.756 *
Presence of DH, no. (%)	22 (48.9%)	13 (35.1%)	0.152 ^†^
IOP parameters			
Baseline IOP, mmHg	14.42 ± 2.73	15.11 ± 3.12	0.292 *
IOP fluctuation during follow-up, mmHg	1.84 ± 2.99	2.72 ± 2.53	0.160 *
OCT parameters			
Baseline rim area, mm^2^	0.84 ± 0.24	0.92 ± 0.27	0.212 *
Slope of rim area, mm^2^/y	−0.08 ± 0.13	−0.07 ± 0.11	0.768 *
Baseline cup volume, mm^3^	0.42 ± 0.25	0.48 ± 0.29	0.350 *
Slope of cup volume, mm^3^/y	0.06 ± 0.06	0.03 ± 0.63	**0.010** *
Baseline average pRNFL thickness, μm	74.40 ± 12.89	75.59 ± 14.47	0.674 *
Slope of RNFL thickness, μm/y	−0.58 ± 0.56	−0.58 ± 0.66	0.987 *
Baseline average mGC/IPL thickness, μm	69.17 ± 9.25	73.18 ± 9.94	0.063 *
Slope of mGC/IPL thickness	−0.61 ± 1.46	−0.57 ± 0.73	0.890 *
VF parameters			
Baseline MD, dB	−5.30 ± 5.25	−4.41 ± 5.34	0.446 *
Slope of MD, dB/y	−0.33 ± 1.37	−0.03 ± 0.53	0.219 *
Baseline PSD, dB	5.96 ± 4.64	4.23 ± 4.27	0.085 *
Slope of PSD, dB/y	0.97 ± 3.31	0.25 ± 0.48	0.187 *
Central progression, *n* (%)	20 (44.4%)	3 (8.3%)	**<0.001** ^†^
Disc parameters using HRT			
HRT ONH depression, *n* (%)	35 (77.8%)	13 (35.1%)	**<0.001** ^†^
Disc area, mm^2^	2.13 ± 0.49	2.13 ± 0.45	0.941 *
Cup shape measures	−0.11 ± 0.07	−0.10 ± 0.10	0.795 *
Slope of cup shape measures	0.01 ± 0.03	−0.00 ± 0.04	0.077 *
Cup depth	0.62 ± 0.11	0.64 ± 0.16	0.450 *
Slope of cup depth	0.01 ± 0.03	0.00 ± 0.03	0.277 *
Measured ONH parameters			
LCD, μm	431.86 ± 93.74	420.25 ± 115.08	0.696 *
Slope of LCD, μm/y	0.58 ± 3.49	0.55 ± 3.97	0.983 *
LCT, μm	201.09 ± 35.12	221.18 ± 44.68	0.081 *
Slope of LCT, μm/y	−3.08 ± 6.64	−0.68 ± 9.73	0.183 *
Follow-up duration, y	9.40 ± 2.01	9.81 ± 2.73	0.437 *

Beta: odds ratio; CI: confidence interval; DM: diabetes mellitus; HTN: systemic hypertension; OCT: optical coherence tomography; pRNFL: peripapillary retinal nerve fiber layer; mGC/IPL: macular ganglion cell–inner plexiform layer; IOP: intraocular pressure; VF: visual field; MD: mean deviation; PSD: pattern standard deviation; dB: decibel; DH: disc hemorrhage; HRT: Heidelberg retinal tomograph; ONH: optic nerve head; LCD: lamina cribrosa depth; LCT: lamina cribrosa thickness; MvD: microvasculature dropout. Data are mean ± standard deviation unless otherwise indicated. * Student’s *t*-test. ^†^ Chi square test. Data are mean ± standard deviation unless otherwise indicated. Factors with statistical significance are shown in bold.

**Table 8 jcm-13-00204-t008:** Follow-up factors associated with MvD development in 82 glaucoma eyes.

Variables	Univariate		Multivariate	
	Beta (95% CI)	*p* Value	Beta (95% CI)	*p* Value
Presence of DH	1.766 (0.723–4.312)	0.212		
IOP during follow-up	1.009 (0.880–1.158)	0.895		
IOP fluctuation during follow-up	0.886 (0.742–1.057)	0.179		
Slope of rim area	0.567 (0.014–3.253)	0.765		
Slope of cup volume	3.380 (0.479–5.280)	**0.015**	3.460 (0.025–6.480)	0.230
Slope of RNFL thickness	1.006 (0.487–2.077)	0.987		
Slope of mGC/IPL thickness	1.027 (0.706–1.494)	0.888		
Slope of MD	0.757 (0.476–1.203)	0.238		
Slope of PSD	1.764 (0.875–3.558)	0.113		
Central progression	8.800 (2.351–32.945)	**0.001**	5.985 (1.474–24.083)	**0.012**
HRT ONH depression	6.462 (2.439–17.120)	**<0.001**	3.765 (1.301–10.895)	**0.014**
Slope of cup shape measures	6.351 (0.153–12.541)	0.086	2.079 (0.001–5.572)	0.530
Slope of cup depth	1.648 (0.003–3.548)	0.278		
Slope of LCD	1.002 (0.856–1.172)	0.982		
Slope of LCT	0.952 (0.884–1.025)	0.188		
Follow-up duration	0.928 (0.770–1.118)	0.432		

Beta: odds ratio; CI: confidence interval; DM: diabetes mellitus; HTN: systemic hypertension; OCT: optical coherence tomography; pRNFL: peripapillary retinal nerve fiber layer; mGC/IPL: macular ganglion cell–inner plexiform layer; IOP: intraocular pressure; VF: visual field; MD: mean deviation; PSD: pattern standard deviation; dB: decibel; DH: disc hemorrhage; HRT: Heidelberg retinal tomograph; ONH: optic nerve head; LCD: lamina cribrosa depth; LCT: lamina cribrosa thickness; MvD: microvasculature dropout. Data are mean ± standard deviation unless otherwise indicated. Factors with *p* < 0.1 in univariate analysis were included in multivariate analysis. Data are mean ± standard deviation unless otherwise indicated. Factors with statistical significance are shown in bold.

**Table 9 jcm-13-00204-t009:** Baseline factors associated with MvD development in 82 glaucoma eyes.

Variables	Univariate		Multivariate	
	HR (95% CI)	*p* Value	HR (95% CI)	*p* Value
Age at diagnosis	0.962 (0.929–0.997)	**0.032**	0.970 (0.926–1.017)	0.210
Female	2.074 (0.795–5.412)	0.136		
Medication of DM	0	0.999		
Medication of HTN	1.264 (0.470–3.401)	0.643		
Vascular symptoms	1.558 (0.543–4.477)	0.410		
SDNN of heart rate variability test	1.010 (0.989–1.031)	0.351		
Axial length	1.015 (0.939–1.098)	0.703		
Central corneal thickness	0.998 (0.985–1.011)	0.752		
Baseline IOP	0.921 (0.790–1.073)	0.289		
Baseline rim area	0.330 (0.057–1.891)	0.213		
Baseline cup volume	0.458 (0.090–2.326)	0.346		
Baseline average pRNFL thickness	0.993 (0.962–1.026)	0.690		
Baseline average mGC/IPL thickness,	0.956 (0.912–1.003)	0.066	0.996 (0.925–1.072)	0.906
Baseline MD of VF	0.967 (0.888–1.053)	0.442		
Baseline PSD of VF	1.094 (0.986–1.213)	0.089	1.116 (0.946–1.317)	0.192
Disc area	1.036 (0.410–2.620)	0.940		
Cup shape measures	0.506 (0.003–79.153)	0.792		
Cup depth	0.284 (0.011–7.235)	0.446		
LCD	1.001 (0.995–1.007)	0.690		
LCT	0.987 (0.971–1.002)	0.089	0.985 (0.969–1.001)	0.073

Beta: odds ratio; CI: confidence interval; DM: diabetes mellitus; HTN: systemic hypertension; OCT: optical coherence tomography; pRNFL: peripapillary retinal nerve fiber layer; mGC/IPL: macular ganglion cell–inner plexiform layer; IOP: intraocular pressure; VF: visual field; MD: mean deviation; PSD: pattern standard deviation; dB: decibel; DH: disc hemorrhage; HRT: Heidelberg retinal tomograph; LCD: lamina cribrosa depth; LCT: lamina cribrosa thickness; MvD: microvasculature dropout. Factors with statistical significance are shown in bold.

## Data Availability

The data that support the findings of this study are available on request from the corresponding author. The data are not publicly available due to privacy or ethical restrictions.

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
