# Peer review of "Factors Contributing to the Development of Choroidal Microvasculature Dropout in Glaucoma Suspects and Patients with Glaucoma"

_jcm, 2023, doi:10.3390/jcm13010204_

Round 1
Reviewer 1 Report
Comments and Suggestions for Authors
Review of the manuscript “Factors contributing to the development of choroidal microvasculature dropout in glaucoma suspects and patients with glaucoma”
The article is devoted to an urgent topic: the study of the nature of the choriocapillary dropout (MvD) in glaucoma. The advantage of the study is that it uses a database of patients with glaucoma who have been observed since 2009 as a component of the Catholic Medical Center Glaucoma Progression 71 Study (CMC-GPS) and the Catholic Medical Center Glaucoma Suspect Cohort Study 72 (CMC-GSCS).
The article is written in good language, contains a large literature review. A fundamental note to the work is to take into account the area of the MvD in dynamics (see below).
1.Abstract
In the abstract, it is necessary to give the main results corresponding to the purpose of the study and confirming the conclusions, indicate their numerical values and the reliability of the differences between groups with and without MvD.
2.Materials and Methods
Patients groups
Line 72
Please clarify if the glaucoma suspects were not the same as the patients with pre-perimetric glaucoma. Otherwise, they should be treated.
How many patients with NTG were enrolled?
Line 179
According to the authors Mvd was identified based on a dropout width >twofold that of visible juxtapapillary microvessels. If MvD had no quantitative expression, then how was the progression (increase in the area of MvD) taken into account in cases of its initial presence?
3.Results
Line 248
ONH depression: please characterize what does it mean.
Table 1
“Baseline demographics and ocular characteristics of 82 eyes of 82 glaucoma patients and 68 255 eyes of 68 glaucoma suspects”. Could you represent this separately for glaucoma group and for glaucoma suspects?
Line 287-288
According to the authors, vascular symptoms or SDNN in the HRV test were not significantly related to MvD occurrence, and it was assumed that systematic blood flow dysregulation had no effect on the occurrence. However, SDNN in the HRV test were not performed with the provocation tests (see below).
Line 364
In the Representative cases please include the information on IOP and its fluctuation during the follow up in both described cases.
4.Discussion
Line 408-410
According to the authors, MvD is associated with glaucomatous damage, particularly in the central macular region. How can you explain this phenomenon?
Line 454
HRT is a less sensitive method for determining structural progression, so it cannot be argued that MvD was not the cause of the thinning of the SNVS, but their consequence.
Line 462
According to the authors, the occurrence of MvD is explained by mechanical reasons. However, the cold provocation test was not performed, so vascular factors as the reason for the MvD appearance can not be excluded. Please mention this in the limitation of the study and do the reference to the paper “Heart rate variability: the comparison between high tension and normal tension glaucoma”published in EPMA J. 2018 Mar; 9(1): 35–45. This paper describes a considerable increase in the activity of the sympathetic ANS in NTG patients in response to the cold provocation test (CPT). The changes of the basic HRV parameters (SDNN, HF, LF, S, and ARI) after the CPT emphasize a significant difference between HTG and NTG patients.
According to literature, the autonomic dysfunction, especially the decrease of SDNNs, was a predictor of central VF progression in NTG (Park HYL, Park SH, Park CK. Central visual field progression in normal tension glaucoma patients with autonomic dysfunction. Invest Ophthalmol Vis Sci. 2014;55(4):2557–2563). Please, include this in your discussion.
6.Conclusions
The authors are strongly recommended to emphasize the appearance of central visual field defects in the both groups (glaucoma and glaucoma suspects) with MvD and try to explain this phenomenon as this is the most important result of the study.
Author Response
Thank you so much for your detailed review.
I'm also attaching a file that shows the highlighted modifications.
Please see the attachment.
Reply to the review report
1.Abstract
In the abstract, it is necessary to give the main results corresponding to the purpose of the study and confirming the conclusions, indicate their numerical values and the reliability of the differences between groups with and without MvD.
-> We added numerical values in the abstract.
"We aimed to characterize and compare the occurrence of peripapillary microvasculature dropout (MvD) between glaucoma suspects and patients with glaucoma. In addition, the factors related to the development of parapapillary MvD in glaucoma suspects and patients with glaucoma were investigated. Total 150 eyes, 68 eyes of glaucoma suspects and 82 eyes of glaucoma patients were analyzed in this study. Univariate and multivariate logistic regression analyses were used to identify factors associated with MvD development. The classification of glaucoma patients or glaucoma suspects was not significantly associated with MvD development (beta 1.368, 95% CI, 0.718-2.608, P = 0.341). In the regression analysis of glaucoma suspect group, greater axial length (beta 1.520, 95% CI, 1.008-2.291, P = 0.046) and baseline cup volume (beta 3.993, 95% CI, 1.292-12.345, P = 0.035) among the baseline factors and the slope of ganglion cell–inner plexiform layer (GCIPL) thickness (beta 0.027, 95% CI, 0.072-0.851, P = 0.027) and central visual field (VF) progression (beta 7.040, 95% CI, 1.781-16.306, P = 0.014) among follow-up factors were significantly associated with the MvD development. In the glaucoma group, central VF progression (beta 5.985, 95% CI, 1.474-24.083, P = 0.012) and ONH depression (beta 3.765, 95% CI, 1.301-10.895, P = 0.014) among follow-up elements were observed as significant factors and the baseline factor had little relationship. MvD appears not only as a result of the progression of axonal loss of RGC in glaucoma but may also be developed by structural changes and mechanical susceptibility of the ONH associated with baseline characteristics. Analyzing the structural susceptibility of the ONH can predict the occurrence of MvD, which is can be helpful in predicting the progression of glaucoma."
2.Materials and Methods
Patients groups
Line 72
Please clarify if the glaucoma suspects were not the same as the patients with pre-perimetric glaucoma. Otherwise, they should be treated.
->The glaucoma suspect patients included in this study were patients who did not have definite evidence of optic nerve damage, which was difficult to see as preperimetric glaucoma, and were therefore not being treated. We have added relevant phrases to the manuscript.
"Glaucoma suspects were defined as individuals with clinical findings or risk factors linked to the likelihood of developing glaucoma, including a high intraocular pressure (IOP) repeatedly above 21 mmHg and suspected findings of the optic disc (concentric enlargement of the cup-to-disc ratio or vertical cup-to-disc ratio ≥ 0.6) or thinning of the retinal nerve-fiber layer (RNFL), and an open-angle evident on gonioscopy, but with no evidence of optic nerve damage or VF loss. "
How many patients with NTG were enrolled?
->Unfortunately, glaucoma suspects and glaucoma patients were distinguished only by structural changes in the optic nerve and the results of visual field tests, and the subtype of glaucoma was not classified in this study, except for the fact that it is open-angle glaucoma.
Line 179
According to the authors Mvd was identified based on a dropout width >twofold that of visible juxtapapillary microvessels. If MvD had no quantitative expression, then how was the progression (increase in the area of MvD) taken into account in cases of its initial presence?
->That's right. Since it is difficult to express a quantitative increase in MvD, in this study, only newly developed MvD cases were confirmed. In order to see the progression of MvD, it seems that the criteria should be considered in future studies.
3.Results
Line 248
ONH depression: please characterize what does it mean.
"ONH depression indicates changes in the connective tissues of the ONH such as the lamina cribrosa and non-axonal components of the neuroretinal rim and prelaminar tissue. "
Table 1
“Baseline demographics and ocular characteristics of 82 eyes of 82 glaucoma patients and 68 eyes of 68 glaucoma suspects”. Could you represent this separately for glaucoma group and for glaucoma suspects?
Yes. Table was changed, separately for glaucoma and glaucoma suspect group.
Line 287-288
According to the authors, vascular symptoms or SDNN in the HRV test were not significantly related to MvD occurrence, and it was assumed that systematic blood flow dysregulation had no effect on the occurrence. However, SDNN in the HRV test were not performed with the provocation tests (see below).
->mentioned in discussion section. (see below)
Line 364
In the Representative cases please include the information on IOP and its fluctuation during the follow up in both described cases.
->IOP & fluctuation is added in case.
4.Discussion
Line 408-410
According to the authors, MvD is associated with glaucomatous damage, particularly in the central macular region. How can you explain this phenomenon?
-> Although the reason remains to be determined, MvD developed preferentially at the inferotemporal sector, which is close to the neuroretinal rim that receives the papillomacular bundle. Hood et al demonstrated that RNFL defects were seen in the region that extends from the inferior portion of the temporal quadrant to the temporal portion of the inferior quadrant in eyes with inferior parafoveal scotoma within the central 10°. Later, the authors defined this region as the macular vulnerability zone. If the MvD affects the structural or functional integrity, or both, of the prelaminar tissue, the MvD would contribute to the development of parafoveal scotoma because of its location near the macular vulnerability zone.
Reference)
Lee EJ, Kim TW, Kim JA, Kim JA. Central Visual Field Damage and Parapapillary Choroidal Microvasculature Dropout in Primary Open-Angle Glaucoma. Ophthalmology. 2018 Apr;125(4):588-596. doi: 10.1016/j.ophtha.2017.10.036. Epub 2017 Dec 8. PMID: 29224927.
D.C. Hood, A.S. Raza, C.G. de Moraes, et al.Initial arcuate defects within the central 10 degrees in glaucoma. Invest Ophthalmol Vis Sci, 52 (2011), pp. 940-946
>We added this in discussion section.
"Therefore, MvD is associated with glaucomatous damage, particularly in the central macular region. This is consistent with previous studies showing that MvD develops preferentially at the inferotemporal sector, which is close to the neuroretinal rim that receives the papillomacular bundle, and consequently contributes to development of parafoveal scotoma."
Line 454
HRT is a less sensitive method for determining structural progression, so it cannot be argued that MvD was not the cause of the thinning of the SNVS, but their consequence.
->I modified the phrase to reflect the reviewer's opinion. “mainly -> likely to be primarily”
"Although ONH depression caused by HRT could result from both ONH changes and rim loss, our findings indicate that ongoing axonal loss at the ONH was likely to be primarily associated with the development of MvD, which may be a secondary finding accompanying glaucoma progression."
Line 462
According to the authors, the occurrence of MvD is explained by mechanical reasons. However, the cold provocation test was not performed, so vascular factors as the reason for the MvD appearance cannot be excluded.
Please mention this in the limitation of the study and do the reference to the paper “Heart rate variability: the comparison between high tension and normal tension glaucoma” published in EPMA J. 2018 Mar; 9(1): 35–45. This paper describes a considerable increase in the activity of the sympathetic ANS in NTG patients in response to the cold provocation test (CPT). The changes of the basic HRV parameters (SDNN, HF, LF, S, and ARI) after the CPT emphasize a significant difference between HTG and NTG patients. According to literature, the autonomic dysfunction, especially the decrease of SDNNs, was a predictor of central VF progression in NTG (Park HYL, Park SH, Park CK. Central visual field progression in normal tension glaucoma patients with autonomic dysfunction. Invest Ophthalmol Vis Sci. 2014;55(4):2557–2563). Please, include this in your discussion.
->We added this in discussion section.
“Third, the vascular symptoms and HRV test were used in this study to evaluate the autonomic dysfunction, but these results alone may be insufficient to figure out systemic blood flow dysregulation. Kurysheva et al. reported that cold provocation test should be performed to more clearly observe changes in HRV parameters, including SDNN, in glaucoma patients.34 In addition, our previous study also reported that central visual field progression in NTG is associated with an autonomic dysfunction.35 Therefore, it may be difficult to say that the occurrence of MvD associated with central visual field progression is entirely unrelated to blood flow instability.”

Reviewer 2 Report
Comments and Suggestions for Authors
The manuscript presents a study on the retinal microvasculature in glaucoma and glaucoma-suspects. It is a part of a larger study which has been run since 2009. The exams were taken yearly which provided a longitudinal observation.
MvD was observed using OCT-A device. The examinations included also visual field, confocal microscopy of optic nerve head (ONH). This additional data enrichened the study, but seems to lead the reader away from the main aim and outcomes of the study.
The study lead to conclusions that are important clinically: MvD might be used as a predictor of early glaucomatous changes. Also, there are risk factors outlined for MvD appearance in patients with glaucoma and glaucoma suspects. They comprise of e.g. low CCT and larger AL, which comes to no surprise, but also of specific parameters in initial visual field and OCT, such as e.g. baseline MD, slope of rim area.
Threre were no major concerns.
Minor concerns:
- the image of MvD depicted by OCT-A and the picture of its measurement are strongly encouraged
- 8 out of 32 references are self-citations. I would suggest that self-citations percentage is reduced
- the results are shown in tables. However, data from tables 4-5, 2-3 and 7-8-9 overlap. I would suggest to present the results in a more clear form, if possible.
- The abbreviations in the abstract must be explained (MvD , VF, OCT, HRT, OCTA, IOP, GCIPL)
- line 126: HRV, please explain the abbreviation
- (Inclusion criteria) in Materials and Methods, there was 'high' IOP as one of criterion for including glaucoma suspects. Please define 'high' IOP, was it recorded once or periodically?
- (Inclusion criteria) in Materials and Methods, auhtors have listed inclusion criteria for incorporating glaucoma patients. Did the patients have to fulfill all of them or just some? Please explain inclusion criteria more precisely.
- line 89: disk (typo)
- line 269: did developed (typo)
- line 444: 'However, RNFL thickness was not significantly associated with RNFL thickness' - please correct the sentence
Author Response
Thank you so much for your detailed review.
I'm also attaching a file that shows the highlighted modifications.
Please see the attachment.
Reply to the review report
Minor concerns:
- the image of MvD depicted by OCT-A and the picture of its measurement are strongly encouraged
-> Figure 1-B and Figure 2-B showed MvD depicted by OCT-A and measurement of MvD was indicated by yellow-dotted line.
- 8 out of 32 references are self-citations. I would suggest that self-citations percentage is reduced
-> These are essential reference papers in making the contents of the thesis, so they cannot be omitted. In the next paper, we will reduce the inclusion of our research in the reference.
- the results are shown in tables. However, data from tables 4-5, 2-3 and 7-8-9 overlap. I would suggest to present the results in a clearer form, if possible.
-> Similar tables were prepared and explained to increase readability by separating the contents related to the occurrence of MvD in all patients, glaucoma suspect, and glaucoma. Thank you for the proposal, but we would like to keep the table format.
- The abbreviations in the abstract must be explained (MvD, VF, OCT, HRT, OCTA, IOP, GCIPL)
-> In order to comply with the Abstract section's word counts regulations, the Abstract contents were modified somewhat, and the description of abbreviation was added.
- line 126: HRV, please explain the abbreviation : HRV (Heart rate variability) assessment
- (Inclusion criteria) in Materials and Methods, there was 'high' IOP as one of criterion for including glaucoma suspects. Please define 'high' IOP, was it recorded once or periodically?
-> High IOP means IOP measured repeatedly (at least 2 times) above 21 mmHg. We have added a relevant phrase.
- (Inclusion criteria) in Materials and Methods, authors have listed inclusion criteria for incorporating glaucoma patients. Did the patients have to fulfill all of them or just some? Please explain inclusion criteria more precisely.
-> The glaucoma patient in this study was defined by the presence of glaucomatous disc, abnormal VF, and open angle in gonioscopy. Included patients have to fulfill all of them. Relevant context was modified.
- line 89: disk (typo) -> modified.
- line 269: did developed (typo) -> modified
- line 444: 'However, RNFL thickness was not significantly associated with RNFL thickness' - please correct the sentence -> modified
